# Influence of Hydrothermal Aging under Two Typical Adhesives on the Failure of BFRP Single Lap Joint

**DOI:** 10.3390/polym14091721

**Published:** 2022-04-22

**Authors:** Yisa Fan, Zhen Liu, Gejin Zhao, Jigao Liu, Yahui Liu, Linjian Shangguan

**Affiliations:** School of Mechanical Engineering, North China University of Water Resources and Electric Power, Zhengzhou 450045, China; fanyisa123@163.com (Y.F.); lz17335585447@163.com (Z.L.); zgj131480@163.com (G.Z.); liujigaokd@163.com (J.L.); liuyahui@ncwu.edu.cn (Y.L.)

**Keywords:** adhesive bonding, basalt fiber reinforced plastics, hydrothermal aging, single-lap joints

## Abstract

Facing increasingly serious resource crises, energy conservation is becoming the development trend of various delivery vehicles, and lightweight is an important way to achieve energy conservation. In this paper, the basalt fiber-reinforced resin composite material (BFRP) was selected to study the effect of its bonding structure, and it was used to make BFRP-BFRP joints. Two adhesives, Araldite^®^2012 and Araldite^®^2015, were used to make single-lap joints and dumbbell-shaped specimens. Aging environments of 80 °C/95% RH and 80 °C/pure water were used for the 0-day (unageing), 10-day, 20-day, and 30-day aging tests, respectively. According to Fick’s second law, the moisture absorption change model of two adhesives was established, and it was found that the water absorption process could be divided into two stages, which explains the precipitation of water molecules and the reaction of water molecules with functional groups. The maximum average failure load and load-displacement curves under different environments and different joints were obtained by using the electronic universal tensile machine, and the exposure time was more important than the effect of humidity. At the same time, the change law of failure strength and ductility were analyzed. The change of T_g_ (Glass transition temperature) was analyzed by differential scanning calorimetry (DSC) equipment, and the results showed that molecular chain rupture was the reason for the decrease of T_g_. It could be seen from the joint failure mode distribution that Araldite^®^2012 adhesive was easily affected by the environment, and the joint of Araldite^®^2015 adhesive was affected by the combined effect of the adhesive and BFRP.

## 1. Introduction

In recent years, due to the increasingly serious shortage of resources, automobile lightweight has become the mainstream of the world’s automobile development. The lightweight body is mainly optimized from the three aspects of body structure, materials and processes to improve the strength and reduce the curb weight of the vehicle [1,2]. Basalt fiber composite material has the advantages of high strength, low density, good corrosion resistance, and low cost [3,4,5]. The fiber composite material has a high degree of flexibility and processing freedom, and its structural design is more flexible and changeable. Basalt fiber-reinforced resin matrix (BFRP) as a new type of composite material has very broad prospects in applications of machinery manufacturing, aviation and the automobile industry [6,7]. The application of various new materials has also produced connection problems. As a new type of connection process, the bonding process can effectively avoid the local stress concentration problems caused by traditional connection processes (such as welding, riveting, and bolting), and has the advantages of large bearing area, uniform stress distribution, and fatigue resistance [8,9].

In the fields of automobiles, high-speed railways and aviation, adhesive joints are widely used due to the need to connect many different materials. In these industries, composite materials are inevitably exposed to complex environments during long-term use. In terms of temperature, the influence of thermal effects on the mechanical properties of the adhesive is related to the ambient temperature and the glass transition temperature of the adhesive. Na et al. [10] used epoxy resin adhesive to study the effect of temperature on the mechanical properties of BFRP/aluminum alloy bonded joints. It was found that as the temperature increased, the tensile strength and Young’s modulus of the joint decreased, while the tensile strain increased. The closer to the glass transition temperature, the more significant the impact on the mechanical properties. In terms of chemical analysis of bonding materials, Galvez et al. [11] conducted DSC (differential scanning calorimetry) and FTIR (Fourier transform infrared spectroscopy) on the adhesive and BFRP after high temperature aging. The results showed that the adhesive underwent post-curing and oxidation reactions in a high temperature environment, and the glass transition temperature (T_g_) increased. BFRP underwent thermal decomposition and oxidation reactions at high temperatures, and T_g_ decreased. Banea et al. [12] studied the stress-strain properties of adhesive dumbbell specimens at room temperature, 40 °C and 80 °C. The results showed that as the temperature increased, the Young’s modulus and failure strength of the adhesive decreased, while the failure strain increased, which was caused by the increase in the toughness of the adhesive as the temperature increased. Akderya et al. [13] studied the effect of thermal aging on the tensile properties of fiber/epoxy composites at three different temperatures (18 °C, 25 °C and 75 °C) for single-lap joints. The results showed that thermal aging at 18 °C increased the bearing capacity of the joint, but as the aging temperature increased, the mechanical properties showed a downward trend.

Regarding humidity, moisture can diffuse into the molecular chain of the polymer. This moisture absorption can be largely attributed to the specific functional groups with polar properties in the cured epoxy resin that have an affinity for moisture [14]. Bowditch [15] described the effect of water on bonded joints. For non-water-absorbent substrates such as metals, water was realized by entering the interface area between the adhesive and the adhesive/substrate. The typical failure mechanism was the hydrolysis and rupture of the bonds at the interface, causing the adhesive to shift. Due to the complex application environment of composite materials, in addition to water immersion, salt immersion and alkaline immersion environments have also been extensively studied. Lu et al. [16] studied the effects of thermal aging, water immersion, and alkaline immersion on the mechanical properties of BFRP plates. The results showed that, in water or alkaline solutions, the tensile strength and interlaminar shear strength of the heat-aged BFRP board sample decreased significantly, while the tensile modulus decreased only slightly. The higher the heat aging and immersion temperature, the more severe the degradation of tensile strength and interlaminar shear strength. Other studies have also shown that BFRP has better alkali resistance than GFRP [17]. Li et al. [18] studied the effects of seawater aging at different temperatures and concentrations on the static/dynamic mechanical properties of carbon fiber reinforced polymer composites. It was found that when the aging time was fixed, the moisture absorption increased with the increase of temperature, but it was not sensitive to the change of NaCl concentration. For the analysis of the moisture absorption characteristics of materials, the most common model is Fick’s law. It assumes that the amount of water absorption is proportional to the concentration gradient in the material. As the aging time increases, the amount of water absorption gradually increases until it reaches saturation.

Temperature and humidity have two different effects on adhesive joints. When the two conditions work together, the adhesive joint will further deteriorate. This phenomenon is called the damp heat aging effect. In the case of high temperature and high humidity, a higher temperature will increase the rate of the chemical reaction and the diffusion coefficient [19], promote more water diffusion, and further cause aging and degradation of the joint, thereby affecting the mechanical properties of the joint. Therefore, it is very important to study the degradation mechanism of composite materials under long-term use. Barbosa et al. [20] studied the influence of moisture absorption on the physical and chemical properties of brittle epoxy resins at different temperatures below the glass transition temperature. They observed that the addition in temperature increased the diffusion rate of water, and as the absorption of water increased, the mechanical properties of the joint deteriorated; this phenomenon is more pronounced at higher temperatures. Zhang et al. [21] studied the shear strength of adhesive joints between thin aluminum and steel substrates at different temperatures and with or without humidity. The following conclusions are drawn that for indoor humidity and various temperatures, there was almost no change in shear strength after 60 days of exposure, but when high temperature and high humidity (90% relative humidity) were combined, the shear strength was significantly affected. Liu et al. [22] studied the effect of high temperature curing epoxy adhesive on the tensile strength of joints under different aging environments. It was observed that the tensile strength decreased by about 31.3% after moisture absorption, and decreased by 27.7% and 54.4% under high temperature (95 °C)/dry environment and high temperature (95 °C)/humid environment, respectively. Zhong et al. [23] comparatively studied the damp-heat durability of glass and carbon fiber reinforced composites, placing GFRP and CFRP composites in an ordinary water bath at 80 °C until they reached water saturation. The study found that compared with CFRP composite materials, GFRP composite materials were more susceptible to humid and hot environments. Ray et al. [24] also obtained consistent experimental results. The decrease in the strength of glass fiber after aging was the main reason for the decrease in the performance of glass fiber reinforced composites.

The damp-heat aging of fiber-reinforced composite materials is the process of material performance degradation under the combined effects of moisture absorption, temperature, time and external stress. The effects of hot and humid environments on composite materials are usually divided into reversible effects (mainly due to moisture) and irreversible effects [25]. Generally speaking, the absorbed water can be removed by drying the material, thereby eliminating the reversible effects, but the reduction of interface strength, the appearance of microstructure defects, and the hydrolysis of the polymer matrix are all irreversible [26]. For example, moisture absorption or cyclic absorption and desorption under high temperature and high humidity will cause voids and/or microcracks in the epoxy resin. Zheng et al. [27] explored the failure mechanism of CFRP/Al joints after damp-heat aging and degradation through mechanical tests. It was found that the strength of the single lap joint showed a non-monotonic change in trend that first decreased and then rose with the damp heat aging time. This was because the post-curing of the adhesive made the adhesive layer harder and the joint’s load-bearing capacity increased. Guo et al. [28] used a multi-scale finite element method to explore the damp and heat aging behavior of carbon fiber reinforced epoxy composites, including the study of aging laws, moisture absorption, and residual stress. The aging of FRP boards is due to the fact that the coefficient of wet expansion (CME) and coefficient of thermal expansion (CTE) of the matrix are higher than that of the fiber. The obvious difference in expansion aggravates the stress concentration and causes microcracks in the interface area [29].

Most existing research is aimed at traditional metal materials and glass fiber reinforced plastics and CFRP substrates, while there is relatively little research on the bonding joints of BFRP substrates. Therefore, this article used epoxy adhesives Araldite^®^2015 and Araldite^®^2012 to conduct experimental research on the durability of experimental specimens under different damp and heat aging conditions. The adhesive bodies (dumbbell specimens) were tested for water absorption under the two environments of 80 °C/95% RH and 80 °C/pure water, and they were analyzed by Fick’s law. The damp and heat aging experiments of the BFRP-BFRP single lap joints under the above two environments were carried out, the load-displacement curves after different aging times (0 days, 10 days, 20 days, 30 days) were obtained, and the changes in the failure displacement and the average maximum failure strength were analyzed. The differential scanning calorimetry (DSC) method was used to study the chemical properties of epoxy adhesives during damp heat. Through macroscopic analysis and scanning electron microscopy (SEM), the failure mode analysis of the corresponding fracture was carried out.

## 2. Experimental Process

### 2.1. Materials

The aim of this paper was to study the aging process under damp and heat aging conditions by making BFRP-BFRP single lap joints and dumbbell-shaped specimens. Two structural adhesives, Araldite^®^2012 and Araldite^®^2015 (Huntsman Advanced Materials), were used to prepare test pieces for the experiment. For the sake of simplicity, Araldite^®^2015 is called A2015, and Araldite^®^2012 is called A2012, as shown in Table 1. Both adhesives are two-component (epoxy resin + curing agent, the ratio is 1:1) epoxy adhesives. A2015 is a tough adhesive with very high lap shear, peel strength and good dynamic load resistance. A2012 is a brittle adhesive that can withstand larger loads, and is resistant to aging, fatigue and corrosion. The specific parameters of the two adhesives are shown in Table 2 (provided by the manufacturer).

The thickness of the BFRP sheet (Jilin Zhongdao Technology Co., Ltd., Baicheng, China) is 2 mm. The sheet is made of plain weave and unidirectional prepreg. The surface density of plain weave and unidirectional fabric is 300 g/m^2^. Basalt fiber (Jilin Tongxin Basalt Fiber Technology Co., Ltd., Tonghua, China) adopts the layup method of [(0/90)0/90/0/90/(0/90)], and the single fiber diameter is 11μm. The composite molding resin (Shanghai Huibo New Material Technology Co., Ltd., Shanghai, China) is composed of GT-807A (epoxy resin) and GT-807B (curing agent) with a ratio (mass ratio) of 100:20. It is a flame-retardant low-viscosity epoxy resin. The resin in the BFRP board is basically 30–35%, and the fiber accounts for 65–70%. The main performance parameters of BFRP are shown in Table 3 (provided by the manufacturer).

### 2.2. Specimen Design and Production

According to the ISO4587 [30] standard, BFRP-BFRP single lap joints were prepared. The bonding joints were made in a dust-free environment with a temperature of (25 ± 2) °C and a humidity of (50 ± 5)%. In order to improve the accuracy of the experiment, all test pieces adopted a unified, standard process bonding process. Proper surface treatment was performed before bonding to obtain bonded joints with good mechanical properties [30]. First, acetone was used to wipe the surface of the bonded BFRP area to remove dust and grease. After drying for 15 min, a special two-component glue gun was used to apply glue to ensure that the adhesive was evenly mixed. Glass beads with a diameter of 0.1 mm were used to control the thickness of the glue layer. Then, the fixation of the test piece on the special bonding fixture was completed (Figure 1). After curing for 8 h at room temperature, the specimen was removed from the fixture and placed in a drying oven (Changge Mingtu Machinery Equipment Co., Ltd., Changge, China) at 30 °C for further curing. After 8 h, the specimen was taken out and put in room temperature to complete the production of the bonded joint. The finished single-lap joint test piece is shown in Figure 2.

The production of dumbbell-shaped adhesive refers to the standard “Determination of Plastic Tensile Properties” ISO527 [31]. A forming mold (Figure 3) was used to make dumbbell-shaped specimens. A two-component glue gun was used to evenly fill the adhesive in the mold. During the filling process, it was necessary to prevent bubbles from appearing in the filled dumbbell-shaped adhesive carefully. After filling, the top plate was aligned with the bottom plate and screws were installed in the peripheral holes to apply a certain pressure to form the shape. Three specimens were made for each adhesive. The size of the dumbbell-shaped specimen after curing was 150 mm × 20 mm, and the thickness was 2 mm, as shown in Figure 4.

### 2.3. Test Method

#### 2.3.1. Water Absorption Test

During the damp-heat aging experiment, an analytical balance with an accuracy of 0.1 mg was used to periodically measure the quality of the BFRP board and dumbbell-shaped adhesive specimens. In the process of measuring the weight of the test piece, the test piece was taken out of the test box and gently wiped with absorbent paper to remove the surface moisture, and then the weight was measured on the high-precision electronic scale. The entire weighing process required wearing disposable gloves and testing quickly to avoid interference from external contaminants and evaporation of water. The water absorption $M_{t}$ is expressed by the following formula:(1)$$M_{t}=\frac{W_{t}-W_{0}}{W_{0}}\times 100\%$$
where $W_{t}$ represents the quality of the sample at time t, and $W_{0}$ represents the quality of the sample at the beginning.

#### 2.3.2. DSC Analysis

The glass transition temperature (T_g_) is an important physical property of amorphous polymers. It is the temperature at which the adhesive transitions between the glass state and the high elastic state. In the actual application process, the glass transition temperature of the adhesive is a key parameter to characterize the strength and performance of the adhesive at a given temperature. In order to explore the influence of the adhesive T_g_ in a hot and humid environment, the thermal properties of the adhesive were measured with a differential scanning calorimetry (Mettler Toledo, DSC3+, Zurich, Switzerland). The experiment was carried out in nitrogen, the ambient temperature was −70 °C–150 °C, and the temperature change rate was 5 °C/min. The adhesive sample was obtained from the failure section of the joint and was about 5 mg. The heating process was repeated twice for each test, the first time to remove the influence of the thermal history, and the second heating process was taken as the final result.

#### 2.3.3. Joint Tensile Test

The finished test pieces were grouped according to different aging environments and aging times, and each environment was divided into four groups according to the four experimental times, for a total of eight groups. Due to the discreteness of the results of the damp-heat aging experiment, 5 replicate samples were prepared for each set of experiments. The specimens cured at room temperature were put into a high-low-temperature damp-heat alternating test box (Weiss Equipment Experiment Company, Dongguan, Guangdong) for 0 days, 10 days, 20 days, 30 days and in two 80 °C/95% RH and 80 °C/pure water aging environments, as shown in Figure 5.

After the aging joint was placed in the experimental environment and returned to normal temperature, the joint was tested on the Xinguang universal testing machine (Jinan Xinguang Testing Machine Manufacturing Co., Ltd., Jinan, China) at a loading rate of 2 mm/min to obtain the load-displacement curves of the bonded joints. The maximum failure strength was the average value of valid data. The drawing process of the specimen is shown in Figure 6.

## 3. Results and Discussion

### 3.1. Analysis of Moisture Absorption Behavior

The moisture absorption behavior of adhesives is generally described by Fick’s law. This law explains the relationship between mass transfer flux and concentration gradient in the process of molecular diffusion. At the beginning of moisture absorption, the absorption quality of water increases linearly with the square root of time, and then the absorption rate slows down until equilibrium is reached. The diffusion rate of water in the adhesive can be described by Fick’s first law, as follows:(2)$$F_{x}=-D\frac{dc}{dx}$$

Among them, $F_{x}$ is the water flux, D is the water diffusion coefficient, x is the water diffusion path, and $\frac{dc}{dx}$ is the water concentration gradient along the x direction, and c is the moisture absorption concentration change with time. The moisture absorption concentration c of the adhesive can be described by Fick’s second law, as shown in Formula (3):(3)$$\frac{\partial c}{\partial t}=D\frac{\partial^{2}c}{\partial x^{2}}$$
where D is the diffusion coefficient, c is the concentration of water, and t is the aging time;

The water absorption properties of structural adhesives depend to a large extent on environmental conditions. In general, the saturated absorption capacity increases with the increase of environmental moisture. Viana [32] mentioned that although Fick’s law was more suitable for simulating the diffusion of water in an adhesive above its glass transition temperature, Fick’s diffusion law could also describe the water absorption behavior of most adhesives in the glass state. In Fick’s model, the analytical formula of water diffusion can be expressed as:(4)$$M_{t}=M_{m}\times \left[1-\frac{8}{\pi^{2}}\sum_{n=0}^{\infty }\frac{1}{{\left(2n+1\right)}^{2}}exp\left(-\frac{{\left(2n+1\right)}^{2}\pi^{2}D}{h^{2}}t\right)\right]\;\;$$

Among them, $M_{m}$ is the moisture content when saturated, h is the thickness of the specimen, D is the diffusion coefficient, and t is the moisture absorption time. The diffusion coefficient D can be obtained by the following formula:(5)$$D=\frac{\pi }{16}\times {\left(\frac{h}{M_{t}∕M_{m}}\right)}^{2}\times {\left(\frac{1}{\surd t}\right)}^{2}$$

The moisture absorption characteristics of the adhesive sample are mainly affected by the damp and heat conditions, the thickness of the sample and the aging time [33]. Using MATLAB software, the relationship between the water absorption rate of the dumbbell-shaped specimen under different conditions and the square root of the degradation time t was calculated by Formula (1), and the moisture absorption behavior is fitted by Formulas (4) and (5), as shown in Figure 7. It can be seen from Figure 7 that the data simulated by Fick’s law is basically consistent with the trend of the data measured by the experiment. The water absorption rate shows an approximate square root increase, and finally reaches saturation and stabilizes.

The entire moisture absorption process is divided into two stages: the unsaturated stage and the saturated stage. In the first stage, the dumbbell-shaped specimen has a faster moisture absorption rate, but the speed gradually decreases, and the slope of the curve decreases with time. In the second stage, the moisture absorption is saturated, the saturated water absorption content of A2012 adhesive is about 14.05%, and the saturated water absorption content of A2015 adhesive is about 16.03%. Generally speaking, part of the water absorbed in the first stage enters the adhesive as free water, occupying a certain space, mainly filling pores and cavities, and supplemented by the reaction of adsorbed water molecules with some hydrophilic functional groups. This leads to plasticization of the adhesive [18]. Careful observation of the curve shows that there is a certain difference between the curve simulated by Fick’s law in the first stage and the actual experimental data. It may be due to two reasons. One is the limitation of Fick’s law itself. Fick’s law only assumes that water diffuses into the material and stays in the free volume, while ignoring the reaction of water molecules with certain functional groups in the material [34]. Second, the experimental data are generally accurate to four decimal places. Such accurate data very easily cause certain errors in the measurement process due to irregular measurement. At the same time, air humidity, room temperature fluctuations and airflow speed also cause errors in the measurement data.

Observing Figure 7, it can be seen that A2015 has the largest diffusion coefficient and saturated water absorption, A2012 is the second, and BFRP is the smallest. Among the three, although the data of A2015 are larger than that of A2012, the two are similar. This is because the main components of the two are similar, and both are epoxy resin adhesives. However, the data of BFRP are not only the smallest, but also very different from the two epoxy adhesives. It is clearly shown that BFRP is located at the bottom layer, which is clearly different from the moisture absorption curves of the two epoxy adhesives. The reason for this is that 65–70% of basalt fibers are present in BFRP. These basalt fibers have the characteristics of no or little water absorption.

In summary, the following conclusions can be drawn: A2015 is most affected by moisture, A2012 is more affected by moisture, and BFRP is least affected by moisture.

### 3.2. The Influence of Damp Heat Aging on the Failure Strength of Joints

Through the quasi-static tensile test, the mechanical performance data of the joint under 80 °C/95% RH and 80 °C/Pure Water environment are statistically processed, and the average failure strength of the joint is obtained with the change rule of the aging time, as shown in Figure 8.

Figure 8 shows the effect of damp heat aging on the failure strength of the joint. At a certain temperature (80 °C), it can be observed that the average failure strengths of the two adhesives depend on the humidity environment and exposure time. No matter what kind of humidity environment, for these two adhesives, it can be obviously observed that the failure strength decreases with the extension of aging time. However, compared with the humidity environment, the effect of exposure time is more important. For the same kind of joint, the strength decreases more obviously after 30 days of aging in the 80 °C/pure water environment. For A2015 adhesive joints, the average failure strength of unaged joints is 11.41 Mpa. After 10 days of aging at 80 °C/95% RH, the joint strength drops from 11.41 Mpa to 10.21 Mpa, a decrease of 10.52%. During the aging process of 10–20 days, the rate of failure strength decreases significantly, from 10.21 Mpa to 9.93 Mpa, only 2.74%. In the last 10 days, the joint strength drops from 9.93 Mpa to 8.25 Mpa, a decrease of 16.92%. After aging at 80 °C/pure water for 10, 20, and 30 days, the joint strength decreases from 11.41 Mpa to 9.33 Mpa, 8.25 Mpa, and 8.09 Mpa, a decrease of 18.23%, 27.70%, and 29.10%, respectively.

In the two different environments, the joint strength using A2015 adhesive gradually decreases with aging time. Compared with the 95% RH environment, the joint strength decreases more significantly after 30 days of aging in the pure water environment. In 95% RH, the area where the joint strength decreases slowly is 10–20 days, while in a pure water environment, the time period where the joint strength decreases slowly is 20–30 days. For the A2012 adhesive joints, the average failure strength of unaged joints is 15.77 Mpa. After 10 days of aging in a 95% RH environment, the strength of the joints deteriorates significantly, from 15.77 Mpa to 11.99 Mpa, a drop of 23.97%. In 10-20 days, the decline rate of the joint strength is significantly slowed down, dropped from 11.99 Mpa to 11.49 Mpa, only 0.5 Mpa. In the last 10 days, the joint strength drops from 11.49 Mpa to 10.16 Mpa, a decrease of 11.57%. After aging for 10 days, 20 days, and 30 days in a pure water environment, the joint failure strength decreases by 2.88 Mpa, 5.69 Mpa and 6.09 Mpa, respectively, compared with the unaged group. After 20 days of aging, the joint strength decreases by 36.08%. After the next 10 days, the joint strength decreases slowly, and the 0–20 days curve is almost linearly decreased.

The remaining strength of the two joints in 20–30 days does not change much (A2012 from 36.08% to 38.62%, A2015 from 27.70% to 29.10%), indicating that the aging degree of the joints has basically reached saturation after 20–30 days under the 80 °C/pure water aging environment. It also corresponds to the moisture absorption behavior of the above two joints, that is, the moisture absorption is basically saturated after 20–25 days.

### 3.3. Representative Load-Displacement Curves of Joints in Different Aging Periods

Through the quasi-static tensile experiment, the load-displacement curve are obtained to explore the joint performance of different adhesives during the stretching process, as shown in Figure 9 and Figure 10.

Figure 9 shows the most representative load-displacement curves obtained at different aging stages under the condition of 80 °C/95% RH. It can be obtained from the ordinate in Figure 9 that as the aging time increases, the failure load of the joint shows an obvious downward trend. For the A2012 joint, the joint failure load decreases from 9.62 kN in the control group to 7.80 kN, 7.18 kN, and 5.45 kN after 10, 20, and 30 days of aging, respectively. The failure loads of the A2015 joint after the three stages of aging are reduced from 7.13 kN to 6.70 kN, 6.70 kN to 6.29 kN, and 6.29 kN to 5.78 kN, respectively. From the abscissa, the maximum displacement of the joint when fracture occurs can be observed. Compared with the failure load, the displacement of the two joints also shows a downward trend with the aging time; the slope of the load-displacement curve corresponds to the stiffness of the bonded joint. For the A2012 adhesive joint, the slope of each curve does not change significantly during the entire damp-heat aging stage, so the stiffness of the A2012 joint is basically not affected in this environment. However, the maximum stretch of the unaged curve is much greater than the other three aging times. For the A2015 joint, it can be clearly seen that its stiffness first rises and then falls in this environment. Compared with the aging group, the stiffness of the unaged group is the lowest, that is, the ability of the aging joint to resist elastic deformation is increased.

Figure 10a shows the most representative load-displacement curves of different aging cycles obtained using A2012 adhesive joints in an environment of 80 °C/100% RH. After 10, 20, and 30 days of aging, the failure loads are 9.10 kN, 7.49 kN, and 6.78 kN, respectively, which decrease by 2.53%, 19.78%, and 27.31% compared with the unaged group. From the slope of the figure, the stiffness of the joint increases after 10 days of aging, and the stiffness of the joint remains basically unchanged after the aging time reaches 20 days. After 30 days of aging, the stiffness of the joint increases slightly compared with that of the unaged one, but not obviously.

Figure 10b shows the most representative load-displacement curves of different aging cycles obtained using the A2015 adhesive joint in the 80 °C/100% RH environment. The performance degradation of the joint after 10 days of aging is very obvious. Compared with the unaged group, the failure load is reduced from 7.13 kN to 5.86 kN, and the maximum displacement at break is reduced from 2.81 mm to 1.69 mm. After 20 days of aging, the joints continue to degrade, but the failure load is not significantly reduced (9.47%). For the joints aged 30 days, it can be seen from the figure that the peak failure load remains basically unchanged (1.40%), and the fracture displacement and stiffness are very close to the joints aged 20 days. Therefore, the authors believe that under an 80 °C/pure water environment, the main aging process of A2015 adhesive joints takes place within 10–20 days.

### 3.4. DSC Analysis Results

Through the differential calorimetric scanning method, the representative DSC curves and average T_g_ values of Araldite^®^2012 and Araldite^®^2015 adhesives were obtained, as shown in Figure 11.

In Figure 11, the DSC curves of the adhesives in the aged and unaged joints are given, and the average value of T_g_ is marked on the curve. The value of T_g_ is related to the change of the DSC slope. The T_g_ of the adhesive is determined by a method similar to [35,36]. Generally speaking, the glass transition temperature (T_g_) is the place where the slope changes the most. The T_g_ value of the A2012 adhesive when it is not aged is 48.58 °C. After 30 days of damp and heat aging, the T_g_ value under the 80 °C/pure water environment is 23.20 °C, a decrease of 52.24%, and the T_g_ value under the 80 °C/95% RH environment is 25.25 °C, which is a decrease of 48.02%. The T_g_ value of A2015 adhesive when it is not aged is 59.25 °C. After 30 days of damp heat aging, the T_g_ value under the 80 °C/pure water environment is 32.24 °C, a decrease of 45.59%; the T_g_ value under the 80 °C/95% RH environment is 41.32 °C, which is a decrease of 30.26%. Generally speaking, a decrease in T_g_ indicates a decrease in molecular weight. Therefore, the reason for the decrease in T_g_ can be explained by the decrease in polymer chains. Hydrolysis can cause polymer chain scission, hard chain and side chain decomposition, molecular weight reduction, free volume increase, and molecular mobility increase [37].

### 3.5. Failure Mode Analysis

The failure modes of single-lap joints at various stages were analyzed through macroscopic images. The typical fracture surfaces of joints exposed to 80 °C/95% RH and 80 °C/pure water during different aging periods are shown in Figure 12.

From Figure 12a, it can be seen that the unaged joints using the A2012 adhesive all show fiber tear failures when they are broken, and with the increase of aging time, the tear failure area (red area) of BFRP composites gradually decreases and the cohesive failure area (yellow area) gradually increases. The degradation of the joint is most obvious after 30 days of aging, and only slight tearing is observed at the edge of the joint, indicating that the 80 °C/95% RH environment mainly reduces the joint strength through the aging of the adhesive. In Figure 12b, the failure mode of the joint in an aging environment of 80 °C/pure water is similar to Figure 12a. The tearing area decreases with the increase of aging time, and the cohesive failure continues to increase. The reason for these problems is that during the aging process, when moisture enters the adhesive or substrate along the cracks, because of the anisotropy of the material itself, the water diffusion presents unevenness, which leads to very irregular shapes.

Figure 12c,d show the failure modes of joints at various stages using the A2015 adhesive, which is significantly different from the A2012 adhesive. In Figure 12c,d, only the unaged specimens show fiber tear failures, and the joints in other aging periods are mixed failures dominated by thin layer cohesion and tearing. In Figure 12c, a thinner adhesive layer can be observed on the fracture surface, indicating that the performance of the A2015 adhesive is significantly affected by damp and heat aging. At the same time, it is observed that part of the fiber tear occurred in the area near the edge of the joint. The fiber filaments (Figure 12(c2)) were pulled out, but the broken surface of 20 and 30 days was not observed, indicating that the degree of fiber tearing gradually decreases with the increase of aging time. The fracture surface morphology of each stage in Figure 12d is basically the same as that of Figure 12c. Both the adhesive and BFRP are polymer materials, and a series of physical and chemical changes occurs under high temperature and humidity. From the observed phenomenon, it can be concluded that the joints using A2015 adhesive are affected by the joint action of the adhesive and BFRP.

In order to explore the microscopic failure mechanism under different modes, the joint failure is analyzed by scanning electron microscopy, as shown in Figure 13. In Figure 13(a2,a3), some potholes are found, as shown by the red ellipse in the figure. The cause of the pits should be the insertion of the fibers under the force of the tearing process. In addition, in Figure 13(a2), it can be clearly observed that the epoxy resin layer on the upper surface of another BFRP is torn off and adhered to this BFRP (yellow wire frame area). This shows that the epoxy resin in the adhesive has good compatibility with the epoxy resin in BFRP, and the two adhere tightly after contact. When there is an external force, the combination of the epoxy adhesive and the BFRP resin layer is closer than the combination of the BFRP resin layer and the fiber layer, resulting in the epoxy adhesive and the BFRP resin layer being combined together, and the fibers in the BFRP are exposed. In Figure 13(b3), it is found that the horizontal fiber is partially broken. The same phenomenon is also reflected in (c1–c3,d2) (in the red box), indicating that the horizontal direction is the main force direction. In the pictures (b3,d1), it is found that there are small and short fiber filaments scattered on the interface (in the blue hexagonal frame), which should be generated when the fiber is broken. At the same time, flake-like dots (in the green triangle frame) are also found scattered on the interface in most of the pictures. Because they have no obvious shape characteristics, it is speculated that they may be torn epoxy resin or fibers or a mixture of both. The majority of Figure 13c,d are bare fiber filaments, indicating that the bonding force between the fiber layer and the resin layer is less than the bonding force between the epoxy adhesive and the resin layer [11]. Considering the cohesive failure of joints, aging leads to the aging of epoxy resin, so the essence of tearing may be internal delamination of epoxy resin. Therefore, it is very likely that there is a very thin epoxy resin layer on the surface of the exposed fiber filaments. Related issues will be discussed below.

In order to explore the composition of the failure surface, the EDX method was used to analyze the element distribution of the failure surface. Table 4 and Figure 14 show the EDX spectra of the two adhesive joints in two different environments under the 30-day aging time. Since the failure section is mainly composed of adhesive, basalt fiber and epoxy resin, its main content is C, O and other elements, as shown in Table 4. The content of the C element is generally close to 80%, the content of the O element is close to 20%, and the two add up to nearly 100%. The C element is mainly carried by the material itself. There are two sources of the O element: one is the material itself, and the other is the hydrolysis under high temperature and high humidity environment. The hydrolysis equation [36] is shown in Equation (6). It is easily shown that the O element in H_2_O is introduced during the hydrolysis process.
(6)$$RCOOR^{\prime }+H_{2}O=RCOOH+R^{\prime }OH$$

It can be seen from the electronic image of Figure 14 that there is a bare fiber layer on the surface. However, according to the composition of the basalt fiber-reinforced composite material (the fiber layer and the resin layer are combined with each other), it is known that there may be a thin layer of epoxy resin on the surface of the torn fiber. It is easy to show that the main components of basalt fiber are silicon dioxide, aluminum oxide, iron oxide and other compounds. Among them, the content of SiO_2_ is the largest, between 45% and 52%. The Si element is an element that does not exist in adhesives and epoxy resins, so Si is used as the characteristic element of basalt fibers. It can be seen from Figure 14 that the distribution of Si elements are sporadic, with a small part concentrated in small blocks or elongated strips. Therefore, it can be seen that after the basalt fiber reinforced composite material is torn, there is still a thin layer of epoxy resin on the surface of the exposed basalt fiber.

## 4. Summary and Conclusions

This paper explored the performance of BFRP-BFRP joints and Araldite^®^2012 and Araldite^®^2015 adhesives in hot and humid environments, provided a reference for them in vehicle service environments, and drew the following conclusions:

(1) Using Fick’s law, the water absorption process is divided into unsaturated and saturated stages. Through saturated water absorption and the diffusion coefficient, it is determined that the A2015 adhesive is most affected by moisture, A2012 is the second- most affected, and BFRP is the least affected;

(2) The average failure strength of the two adhesives depends on the humidity environment and exposure time. No matter what kind of humidity environment, it can be obviously observed that, for these two adhesives, the failure strength decreases with the extension of the aging time. However, compared with the humidity environment, the influence of exposure time is more important;

(3) Under the condition of 80 °C/95% RH, the displacement of the joint when it breaks first decreases and then increases, and the ductility of the A2012 joint in this environment is slightly better than that of the A2015 joint. Compared with the unaged joint, the ability of the aged joint to resist elastic deformation is increased. The most serious moment of joint degradation of the single-lap joint of the A2012 adhesive under the condition of 80 °C/pure water is at 30 days. The A2015 declines rapidly in the first 10 days, and gradually slows down in the next 20 days;

(4) The reason for the decrease of T_g_ can be explained by the decrease of the polymer chain. Hydrolysis can cause polymer chain scission, hard chain and side chain decomposition, and molecular weight reduction;

(5) In the analysis of the joint failure, it was found that the tear area of the A2012 adhesive joint continued to decrease with the increase of aging time, and the cohesive failure increased with time, indicating that the aging environment mainly affected the adhesive. In the A2015 adhesive, it was found that the thin layer cohesive failure and tearing failure occurred at the same time, indicating that the joint using the A2015 adhesive was affected by the combined effect of the adhesive and BFRP;

(6) In the elemental analysis, Si was used as the characteristic element of basalt fiber filaments, and it was found that after the basalt fiber reinforced composite material was torn, there was still a thin epoxy resin layer on the surface of the exposed basalt fiber.

## Figures and Tables

**Figure 1 polymers-14-01721-f001:**
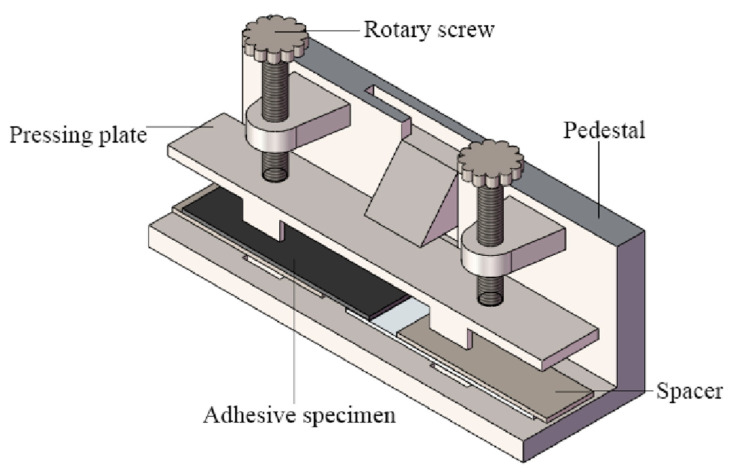
Adhesive fixture of single-lap joint.

**Figure 2 polymers-14-01721-f002:**
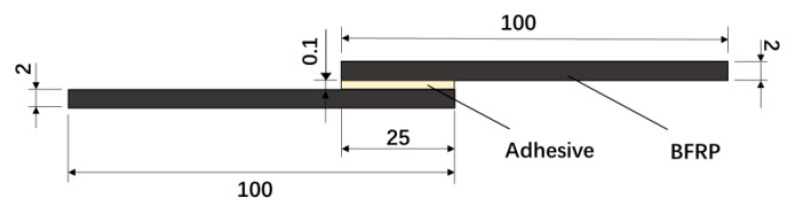
Geometry of single lap adhesive joint with 25 mm width (dimensions in mm).

**Figure 3 polymers-14-01721-f003:**
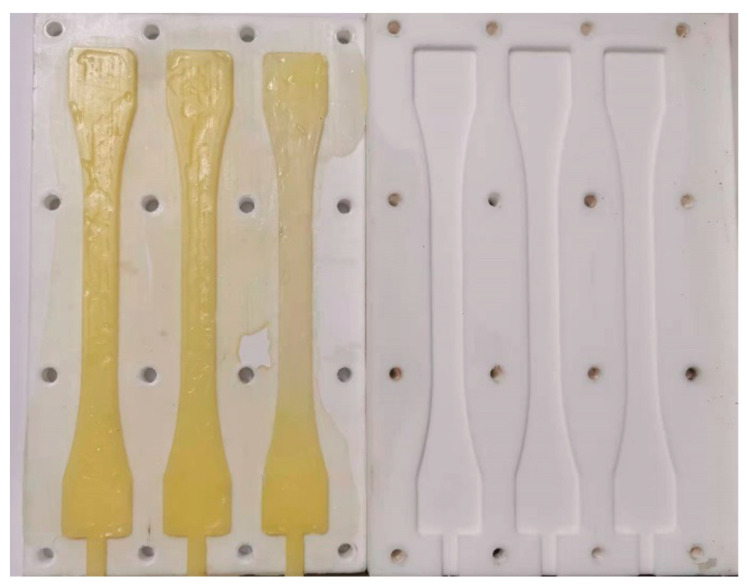
Mould for making dumbbell specimen.

**Figure 4 polymers-14-01721-f004:**
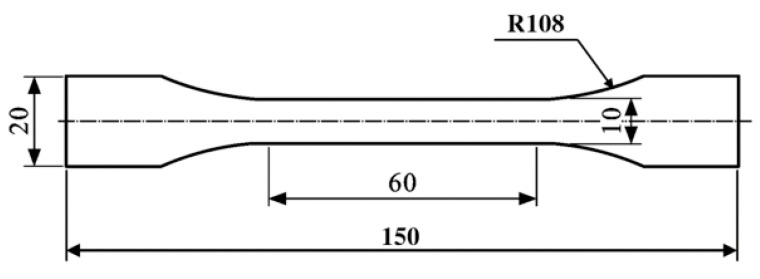
Geometric shape of dumbbell shaped specimen.

**Figure 5 polymers-14-01721-f005:**
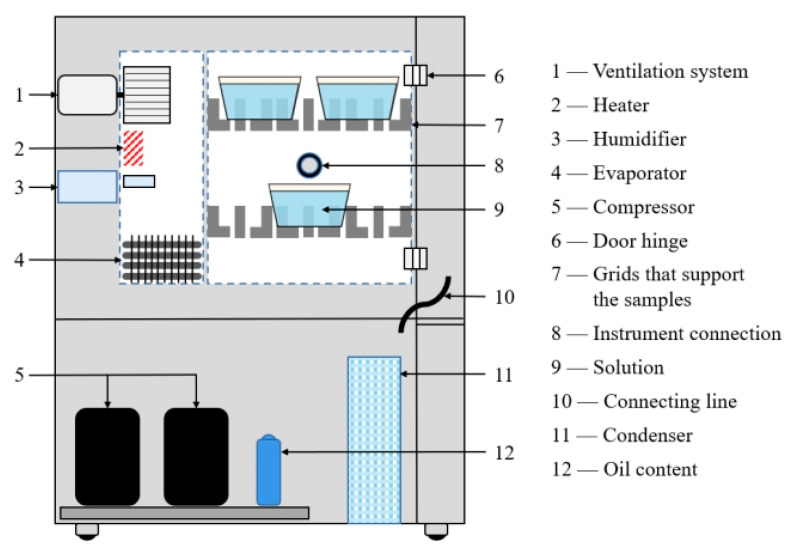
Environmental cabinet principle model.

**Figure 6 polymers-14-01721-f006:**
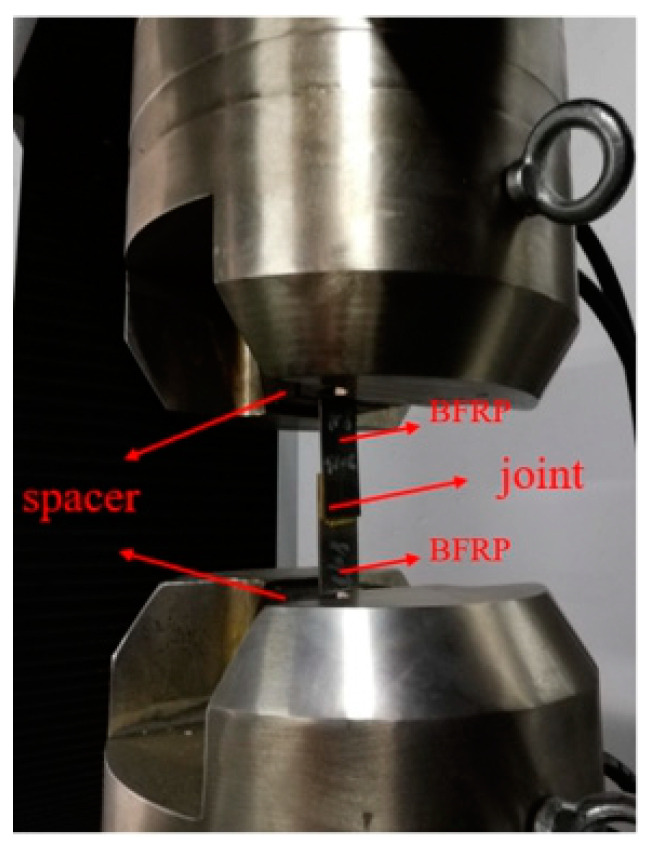
Schematic drawing of the stretching process.

**Figure 7 polymers-14-01721-f007:**
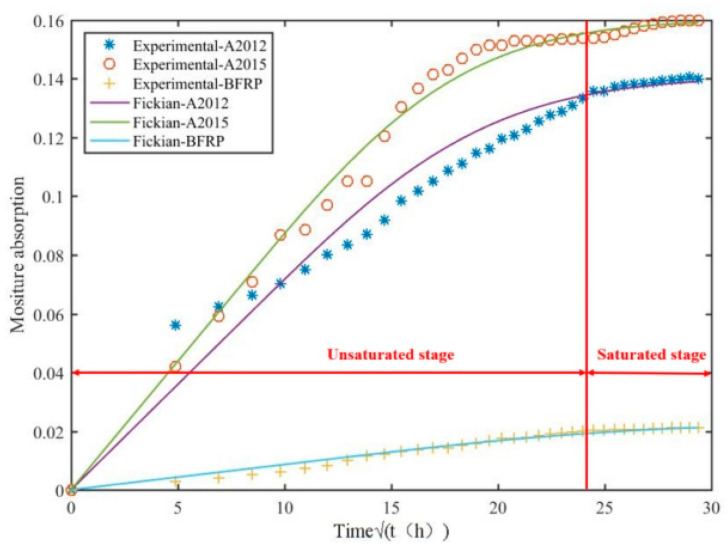
Experimental and analytical moisture absorption curves of the adhesive bulk specimen.

**Figure 8 polymers-14-01721-f008:**
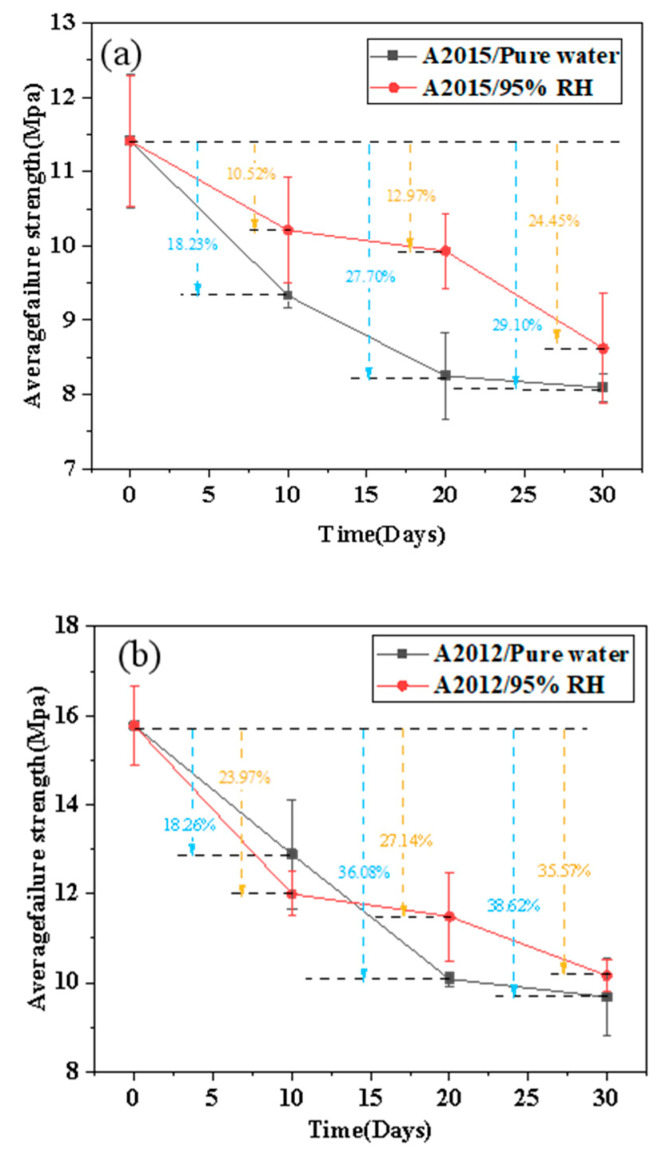
Effect of hygrothermal aging on the strength of single lap joint: (**a**) A2015 and (**b**) A2012.

**Figure 9 polymers-14-01721-f009:**
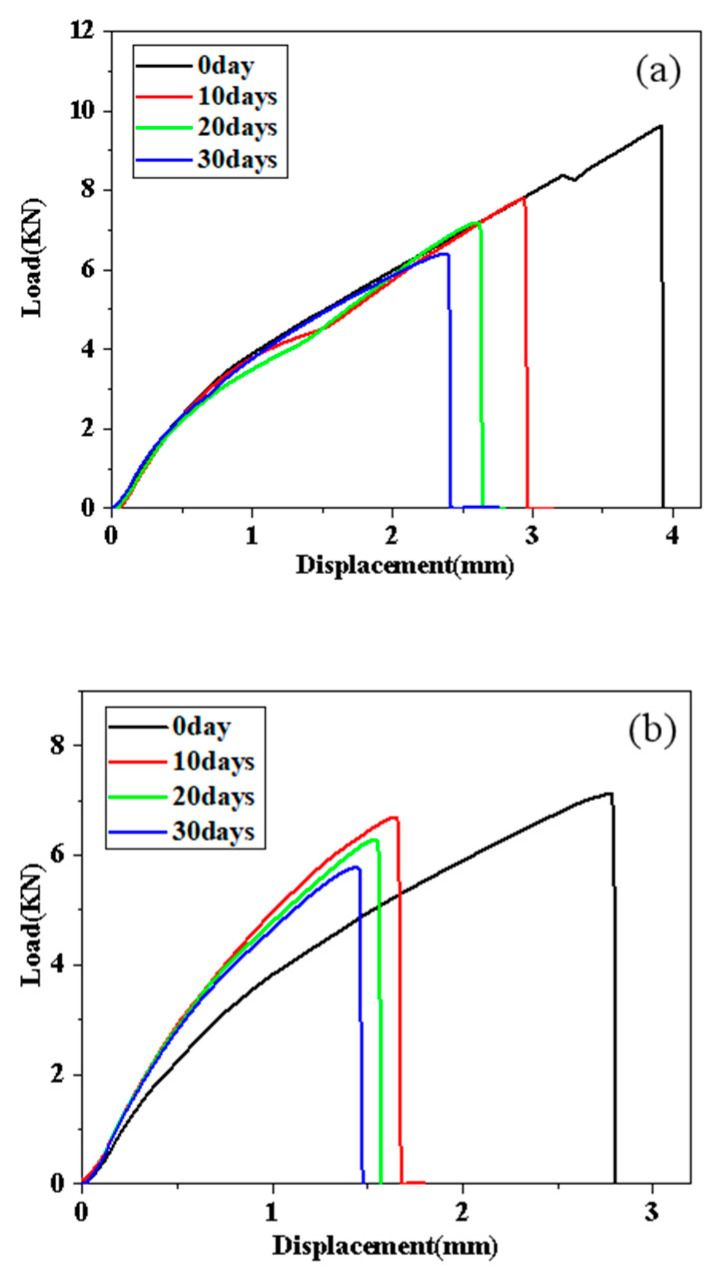
The most representative load displacement curves of different aging stages were obtained: (**a**) A2012-95% RH and (**b**) A2015-95% RH.

**Figure 10 polymers-14-01721-f010:**
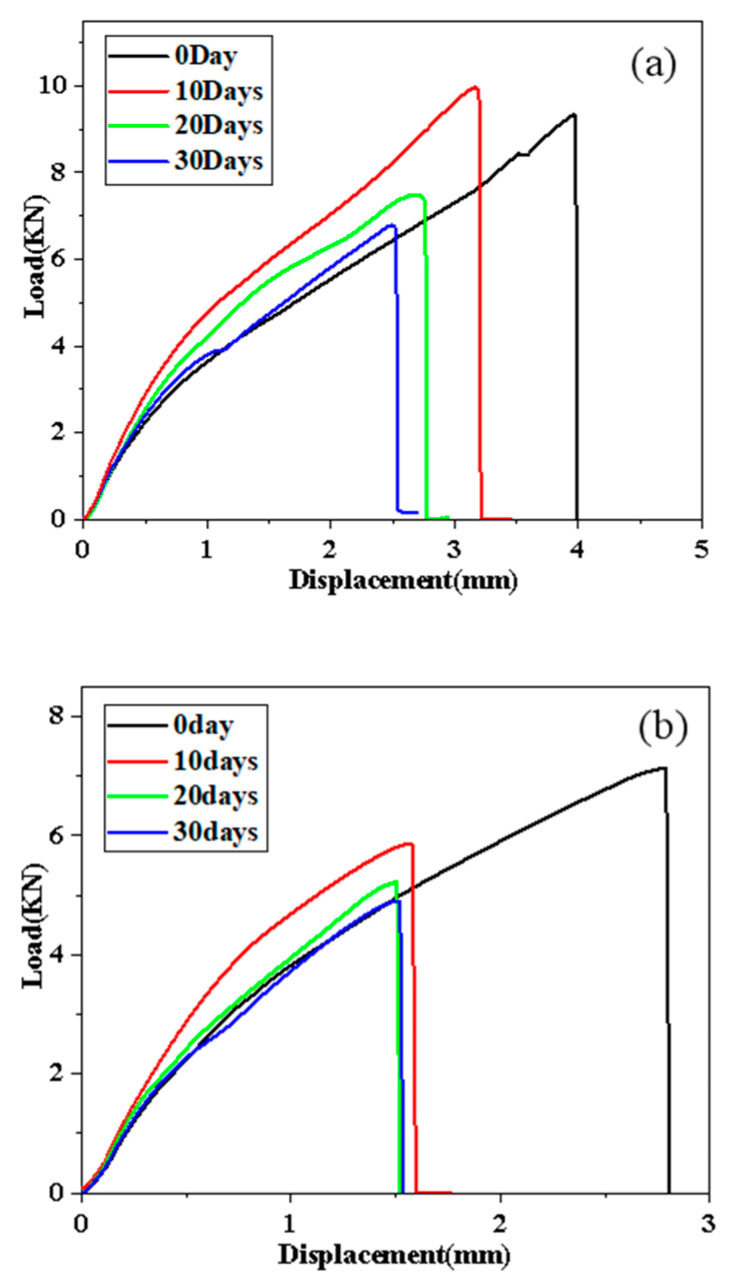
The most representative load displacement curves of different aging stages were obtained: (**a**) A2012-100% RH and (**b**) A2015-100% RH.

**Figure 11 polymers-14-01721-f011:**
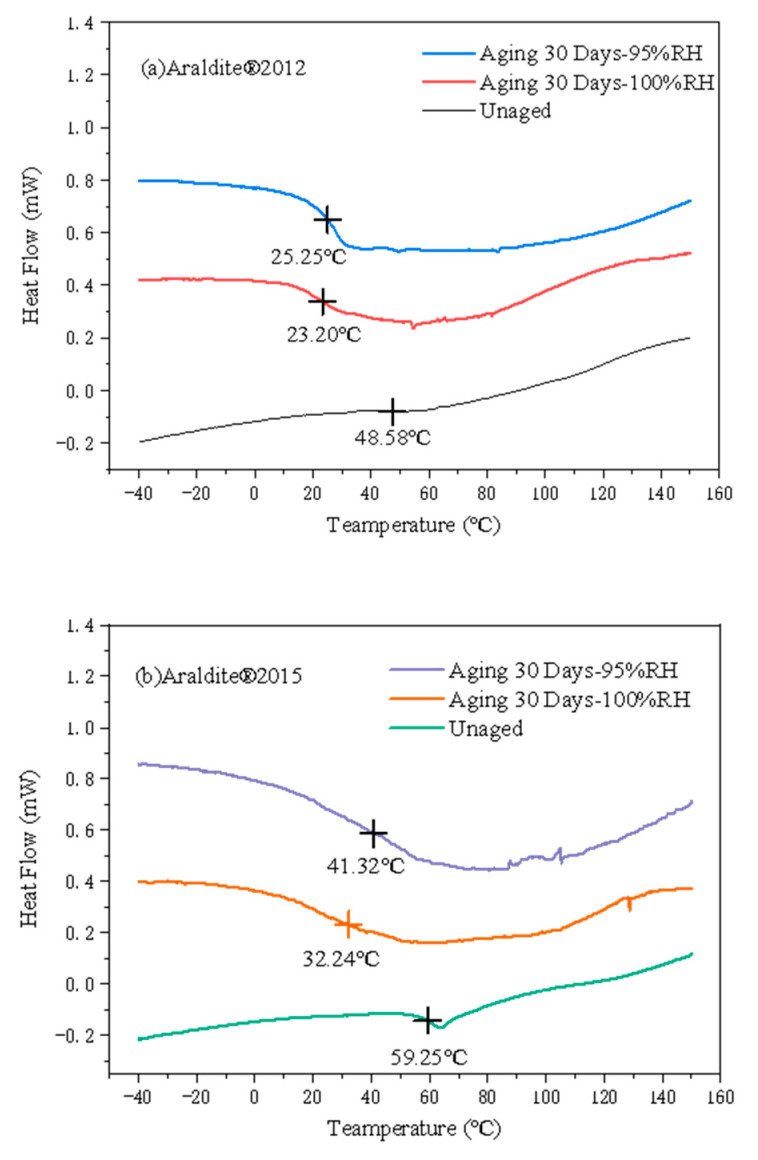
Representative DSC results and average Tg assignment of adhesive: (**a**) A2012 and (**b**) A2015.

**Figure 12 polymers-14-01721-f012:**
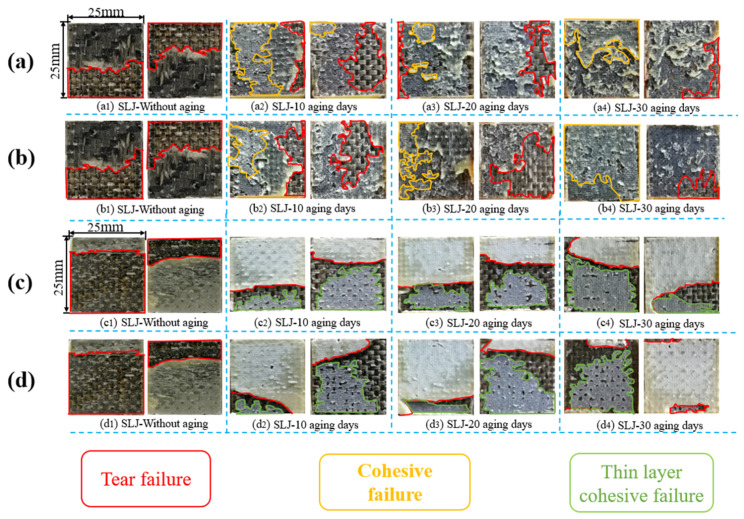
(**a**) The surface of A2012 joints in 80 °C/95% RH; (**b**)the surface of A2012 joints in 80 °C/pure water; (**c**) the surface of A2015 joints in 80 °C/95% RH; (**d**) the surface of A2015 joints in 80 °C/pure water.

**Figure 13 polymers-14-01721-f013:**
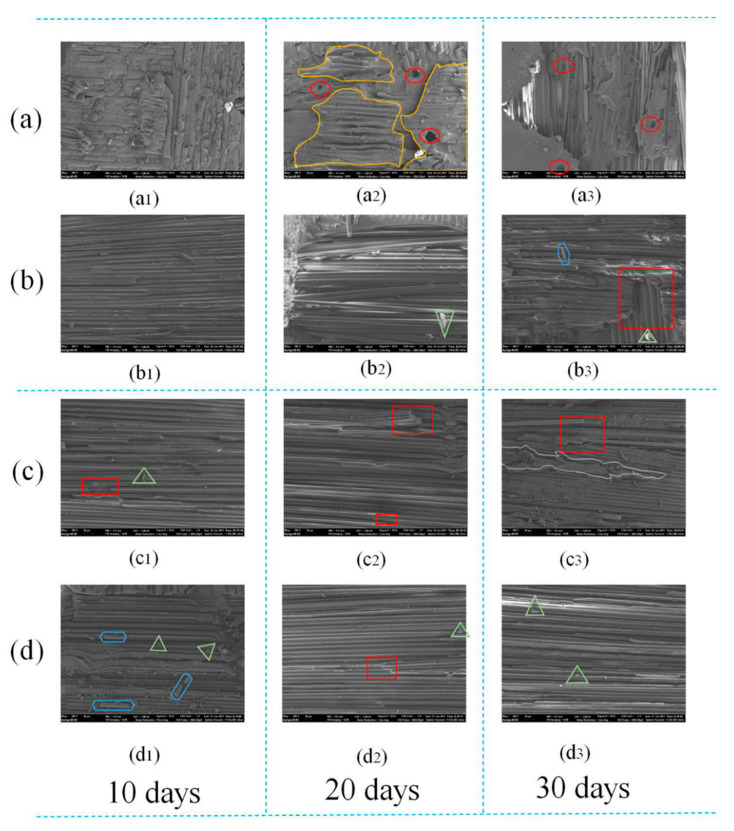
The SEM micrograph of different adhesive joints in different environments: (**a**) A2012 joints in 80 °C/95% RH, (**b**) A2012 joints in 80 °C/pure water, (**c**) A2015 joints in 80 °C/95% RH, and (**d**) A2015 joints in 80 °C/pure water.

**Figure 14 polymers-14-01721-f014:**
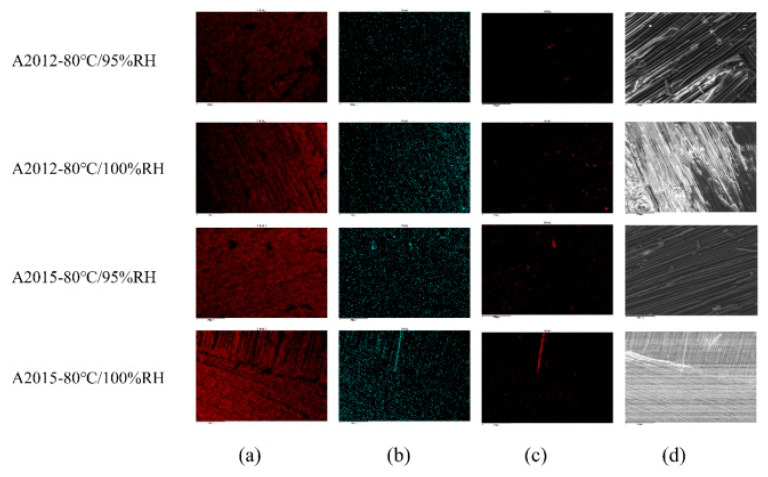
EDX under different environments and adhesives for 30 days: (**a**) C element, (**b**) O element, (**c**) Si element, and (**d**) electronic image.

**Table 1 polymers-14-01721-t001:** Specimen ID and specific materials.

Specimen ID	Materials
A2012	Araldite^®^2012
A2015	Araldite^®^2015

**Table 2 polymers-14-01721-t002:** Material properties of adhesives.

Properties	Araldite^®^2012	Araldite^®^2015
Young’s modulus, E [GPa]	1.65	1.85 ± 0.21
Shear modulus, G [GPa]	0.25	0.56 ± 0.21
Poisson’s ratio	0.38	0.33
Density (kg/m³)	1.18	1.4

**Table 3 polymers-14-01721-t003:** Material properties of BFRP.

Fiber Direction	E_x_/GPa	E_y_/GPa	G_xy_/GPa	V
Unidirectional	105 ± 10	8.5 ± 2	6.5 ± 0.5	0.06
Plan weave	45 ± 5	45 ± 5	3 ± 0.3	0.1

**Table 4 polymers-14-01721-t004:** Different element content in EDX.

	Element	C	O	Si
Type				
A2012-80 °C/95% RH		79.28	17.76	0.14
A2012-80 °C/100% RH		76.97	19.56	0.23
A2015-80 °C/95% RH		78.49	19.75	0.16
A2015-80 °C/100% RH		79.35	18.03	0.17

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
