# Peer review of "Influence of Hydrothermal Aging under Two Typical Adhesives on the Failure of BFRP Single Lap Joint"

_polymers, 2022, doi:10.3390/polym14091721_

Round 1

Reviewer 1 Report

I would like to thank, the editor of the Polymer for trusting me to review the paper “Influence of hydrothermal aging under two typical adhesives 1 on the failure of BFRP single lap joint”

My decision regarding the current work is accepted with minor changes as listed below:

  1. The literature review section need modification as limited papers from 2021 and 2022 are included in this section.
  2. Why the current aging times were selected? Any special reason for this?
  3. What will be the effect of load ratting effect in tensile testing on the tested specimens?
  4. The resolution of Figure 12 is very poor, I will suggest to improve it for readers.
  5. Can the authors include limitations of the current work in terms of dynamic thermal loading?

Reviewer 2 Report

The subject of the work is a strength analysis of the glued joint of elements. The lap joint was tested. Experimental measurements were made to determine the strength of the connection. Accepting cut oars as a sample is not a fully accurate solution. Although such a solution is accepted. Glued joints are tested according to specific standards. The presented work is quite extensive. The authors present a good researcher's technique. The introduction and analysis of the literature are correct and provide a good background for the research. The article is a kind of research report. The presented figures and tables are clear and the descriptions are in the main text. The conclusions formulated are correct.
